# Deep Learning and Procrustes Analysis for Early Dysgraphia Risk Detection with a Tablet Application

**DOI:** 10.3390/life13030598

**Published:** 2023-02-21

**Authors:** Eugenio Lomurno, Linda Greta Dui, Madhurii Gatto, Matteo Bollettino, Matteo Matteucci, Simona Ferrante

**Affiliations:** Department of Electronics, Information and Bioengineering, Politecnico di Milano, 20133 Milan, Italy

**Keywords:** dysgraphia, longitudinal monitoring, early screening, time series embedding, procrustes analysis, deep learning

## Abstract

Dysgraphia is a neurodevelopmental disorder specific to handwriting. Classical diagnosis is based on the evaluation of speed and quality of the final handwritten text: it is therefore delayed as it is conducted only when handwriting is mastered, in addition to being highly language-dependent and not always easily accessible. This work presents a solution able to anticipate dysgraphia screening when handwriting has not been learned yet, in order to prevent negative consequences on the individuals’ academic and daily life. To quantitatively measure handwriting-related characteristics and monitor their evolution over time, we leveraged the Play-Draw-Write iPad application to collect data produced by children from the last year of kindergarten through the second year of elementary school. We developed a meta-model based on deep learning techniques (ensemble techniques and Quasi-SVM) which receives as input raw signals collected after a processing phase based on dimensionality reduction techniques (autoencoder and Time2Vec) and mathematical tools for high-level feature extraction (Procrustes Analysis). The final dysgraphia classifier can identify “at-risk” children with 84.62% Accuracy and 100% Precision more than two years earlier than current diagnostic techniques.

## 1. Introduction

Handwriting is one of humanity’s first major technological revolutions in systems of communication and self-expression. It is a skill typically developed in early childhood [1] and typified by substantial complexity. Cognitive activities related to handwriting include general thinking, creativity, and language comprehension, while, from a motor point of view, several joints between hand and arm are involved, all of which must be precise and synchronous. Despite the increasing digitization of the education system, handwriting remains a central skill in every child’s schooling and learning. In fact, recent studies based on the examination of brain activity through electroencephalography show that children learn and remember better when writing with a pen rather than a keyboard [2]. However, up to 27% of children [3], even if followed in the learning phase, fail to correctly learn handwriting and thus are unable to perform writing, composing, and spelling tasks [4].

Dysgraphia, from the Greek “dys” meaning “impaired” and “graphia” meaning “making letter forms by hand”, is the neurodevelopmental disorder of biological origin in the reproduction of alphabetical and numerical characters. The writing skills of individuals with a dysgraphic condition are below the level expected for their age and cognitive level, despite adequate learning opportunities and the absence of any evident neuropathologies or sensory-motor problems [5]. Dysgraphia may manifest itself in a stand-alone manner, or it may also coexist with other learning difficulties [6], with the possible consequence of being easily confused with other transient disorders, or even not being diagnosed at all [7]. Dysgraphia affects not only the school learning process but also daily life activities, with possible negative emotional and behavioral consequences for children, such as increased anxiety, loss of self-esteem, and risk of precocious school abandonment [6,8]. It is therefore important to monitor children’s handwriting abilities over time, in order to be able to detect handwriting weaknesses as early as possible and to properly intervene and avoid negative consequences [8] on the academic, working and daily individuals’ lives.

Typically, the diagnosis of dysgraphia is made after the second year of primary school, when the process of learning to handwrite should be well established. Consequently, it is currently almost impossible to detect dysgraphia before that age, which inevitably leads to negative consequences on the lives of the most disadvantaged children. In particular, among the weaknesses of current methodologies, one can highlight:The possibility of intervening only after the time when handwriting skills should have been learned, slowing down the diagnosis and making the adoption of countermeasures less effective [9].The strong dependence on the native language.The necessity to depend only on handwritten text, and thus without providing a quantitative assessment of gestural production [10].The requirement for children to reach the clinicians’ office, which may not be easily accessible in terms of costs and waiting time [11].

This work addresses all the current critical issues in dysgraphia diagnosis, developing a solution that can move towards a new approach, that of anticipating dysgraphia screening at an age when handwriting is not mastered yet, starting from symbols drawing. Indeed, since the object of study of dysgraphia are children, a population whose abilities are constantly developing, it is essential to investigate how growth and education influence their writing skills over time and, in particular, how this development process differs in children potentially “at-risk”. This, together with the fact that the disorder of dysgraphia has an intrinsic developmental character, leads to the extreme need for longitudinal monitoring of handwriting skills. Given this rationale, this paper presents an early dysgraphia classifier based on real, quantitative, and longitudinal data, developed through state-of-the-art mathematical tools and deep learning techniques. Raw data collected through the serious games for a tablet [9] were cleaned and subsequently reduced in dimensionality through autoencoder-based architectures. The result of this process was used to calculate outlier distances through Procrustes Analysis, to realize threshold-based classifiers and classification meta-models. The final result is a tool able to identify “at-risk” children in the pre-literate age, and particularly, more than two years earlier than the currently applicable protocol. The performance achieved, i.e., a false positive rate equal to zero and an Accuracy of 84.62%, suggests that the proposed approach represents a reliable tool with great potential utility for clinicians, parents, and teachers.

## 2. Related Works

With the growth of technological development, the assessment of handwriting quality is increasingly being entrusted to digital tools. The advent of tablets is opening up a new world for the quantitative analysis of writing problems: their advantage is that, by recording handwriting in real-time, they allow for a kinematic analysis of the written stroke, and thus consider not only the static and final aspects of handwriting but also its dynamics, which has proven to be fundamental in the analysis of handwriting disorders. The aim of much of the work developed in recent years is to combine the effectiveness of traditional methods based on subjective rules with more general data-driven approaches that exploit statistics and machine learning techniques [4].

### 2.1. State of the Art

In 2013, Accardo et al. [12] examined with a cross-sectional study data on students attending classes from the second grade of primary school to the third grade of secondary school. They were asked to perform four different cursive tasks on a commercial digitizing tablet, using a sheet of lined paper affixed on its surface and an ink pen in order to reproduce a normal “pen and paper” context. Within this study, the authors demonstrated the utility of kinematic writing parameters to study the development and learning of handwriting over time. The results highlighted how these writing parameters could be used to objectify the diagnostic methods, thus quantifying the deficit of hand movement skills in patients of the same age presenting handwriting difficulties.

In 2018, Asselborn et al. [13] proposed a study considering children performing the dysgraphia BHK test from the first to the fifth grade, again through a digital tablet covered with a sheet of paper. By means of a random forest classifier, they discovered that frequency-related features were the most discriminative to predict dysgraphia. Two years later, the same authors contributed to the dysgraphia study with a new paper in which they asked children to perform the BHK test using the Apple Pencil on a specially designed iPad application (Dynamico app v1.0) [4]. In this study, they were not interested in a binary dysgraphia result, but rather a scale of handwriting difficulty. The idea was therefore to calculate how much a child’s score deviated from the average score of children of the same age and gender. Taking advantage of the PCA and k-means algorithms, they first discovered that tilt and static did not appear to be significant in explaining the differences in handwriting between children, as opposed to kinematic and pressure characteristics. Finally, they were able to create threshold values within their test population, resulting in five classes of handwriting difficulty to assess the children.

In 2020, Richard and Serrurier [14] collected data consisting of images of handwritten text and audio recordings, both from children without any diagnosis and from children diagnosed with dyslexia and/or dysgraphia. The aim was to apply classical machine learning techniques (such as Naïve Bayes, Logistic Regression, and Random Forest) to obtain a model capable of identifying one of the two neurodevelopmental disorders. Thanks to its promising results, this research is seen as a further demonstration of how artificial intelligence can effectively help automate and predict the assessment of such disorders.

In the same year, Drotár and Dobeš [15] used a machine learning approach to discriminate between deteriorated handwriting and dysgraphia on different writing tasks performed by children aged between 8 and 15 years. Also in this work, the authors opted for the use of a tablet, this time used with a sheet of paper on which to write with a pen. By comparing different machine learning algorithms, the results presented in the paper showed that machine learning tools could be used to detect dysgraphia, even when dealing with a heterogeneous set of subjects that differ in age, gender, and hand.

In 2021, Deschamps et al. [16] further investigated the possibility of anticipating dysgraphia screening by developing an automated, widely applicable pre-diagnostic tool. By asking more than 500 children from second to fifth grade to perform the French BHK test on a graphic tablet, they collected data consisting of almost a hundred features of the written tracks. By testing different machine learning algorithms, they achieved a satisfactory discriminatory performance, comparable to that of a human examiner, reinforcing the belief that such tools would be of great benefit in the detection of dysgraphia.

In 2022, Ghouse et al. [17] explored many deep learning techniques for classifying dysgraphia in children’s handwritten images. In particular, they used convolutional neural networks to improve the efficiency of traditional manual dysgraphia classification. They focused on mitigating the overfitting problem that characterized previous machine learning-based research, introducing appropriate tuning parameters and achieving promising results.

The results from these studies are promising and useful in a pre-clinical context, but they share a limitation with classical diagnostic tools. In fact, they rely on the analysis of handwritten text, thus requiring to postpone the evaluation when handwriting is learned.

Recently, Dui et al. [11] confirmed the effectiveness of deep learning in the context of dysgraphia prevention even before handwriting is learned. Using serious games from the iPad application Play-Draw-Write, the authors collected data from children in the last year of kindergarten, and through the use of a convolutional neural network called LearNet, they were able to effectively discriminate between “at-risk” and “non-at-risk” children. In addition, the work showed how the Play-Draw-Write application can be complementary to expert observation, thus becoming a valuable aid in identifying areas of difficulty to be strengthened, even without the need for a trained teacher, or even in a remote context. The main limitation of the study was the uncertainty of labeling the risk of dysgraphia. In fact, as the authors pointed out, the risk of grapho-motor retardation was assessed by the subjective and non-clinical judgment of a pool of trained teachers, and therefore highly subject to noise. Hence, a longitudinal observation of the same children until it was possible to test for their real handwriting abilities was necessary.

Further studies on the use of artificial intelligence to analyze and prevent dysgraphia are summarized in the work of Moetesum et al. [18].

### 2.2. Play-Draw-Write Application

The present study is based on Play-Draw-Write, the tablet application developed at the NearLab of Politecnico di Milano University [9]. It was developed in Unity 2018.3.2f1, for an iPad 6 with Apple Pencil 1 characterized by a sampling frequency of 240 Hz for stylus position, pressure, altitude, and azimuth angles. Play-Draw-Write was designed to assess handwriting-related features, starting from symbol drawing instead of word production analysis. It was developed to address the weaknesses of current dysgraphia diagnosis methodologies and therefore to meet the following important requirements [19]:To provide a quantitative assessment of gestural production and thus objective parameters that are typically altered in dysgraphic handwriting.To be sensitive to children’s learning development, in order to discriminate longitudinal improvements, thus allowing to anticipate dysgraphia screening to a pre-literacy age.To be easily usable, independently from the language, and easily accessible for non-clinical users to make the detection of handwriting difficulties accessible to the whole community.To amuse children in order to facilitate repetitive use over time.

The assessment of gestural production characteristics is made through the quantification of three handwriting principles, which are known to be altered in dysgraphic handwriting [20,21]: *isochrony*, *homothety*, and *speed–accuracy trade-off (SAT)*. *Isochrony* is the property according to which the writing speed increases with stroke size, such that the execution time remains approximately constant. *Homothety* is the property according to which the fraction of time spent on each letter, relative to the total time for the entire word, is constant, independently from the stroke size. *SAT* is instead the property according to which the more accurate the task, the longer it takes to accomplish it, and vice versa. Play-Draw-Write investigates handwriting through Serious Games applications, which boost pupils’ engagement, enable continuous monitoring, and allows data acquisition to evaluate the above laws [22].

The serious game categories considered by the present study are Copy and Tunnel Games. Copy Games require to copy symbols on an empty canvas, beginning from a square in Copy Square exercise and then following with a symbols’ sequence, i.e., a circle, a line, and a reversed U, in Copy Sequence exercise. Both Copy Games have three different copy modalities, i.e., spontaneous, big, and small. This game is designed to study *isochrony* and *homothety*. Tunnel Games instead require to pass first through 15 different square-shaped tunnels in Tunnel Square exercise and then through 15 different italic-“ele”-shaped tunnels in the Tunnel Word exercise, in both cases as fast as possible and without crossing the borders. Tunnel Game is designed to study *SAT*. Indeed, the Tunnel Game presents 15 tunnels of varying amplitude (A), i.e., the path length, and width (W) in a random order, to create different Indexes of Difficulty (ID = $\frac{A}{W}$, in particular, 5 for Tunnel Square and 8 for Tunnel Word). IDs represent the difficulty to achieve the tunnel task so that the greater the ID, the more difficult the task [23]. Sample screenshots of the four games considered are shown in Figure 1.

The main advantage of Play-Draw-Write lies in the possibility of being played by children from any country in the world, regardless of the language spoken or the alphabet used. Furthermore, it is a tool that allows for objective assessment of handwriting development, potentially administered in a pre-clinical setting via teleconsultation or remote monitoring. Play-Draw-Write is therefore a technological support that paves the way toward an early dysgraphia detection: it allows the screening of pre-literacy skills to identify potential handwriting difficulties before they arise and offers continuous, accessible, and low-cost monitoring of their evolution, while pupils enjoy playing games [19].

## 3. Method and Experiments

The aim of this research work is to exploit the longitudinal administration of the Play-Draw-Write test to anticipate dysgraphia screening at an age when handwriting has not been mastered yet.

### 3.1. Participants and Protocol

The longitudinal protocol followed to collect data on the children’s handwriting skills involved five acquisitions over three years: the first measurement was taken in February of the last year of kindergarten (2020), while the last one was taken in May of the second grade (2022). During the last time point, children were also asked to perform the BVSCO-2 Test (Batteria per la Valutazione della Scrittura e della Competenza Ortografica) [24], that is, the most widely adopted and standardized test in Italy for detecting handwriting difficulties. The BVSCO-2 Test leveraged for handwriting skills assessment in this work comprises three handwriting exercises:Writing a sequence of letters L and E in cursive, without lifting the pen, for one minute;Repeating the word UNO (ONE, in Italian) in the preferred case, for one minute;Writing numbers as letters, in the preferred case, for one minute.

In each exercise, when the children score is less than two standard deviations below the normative mean of the same age and period of the year, they are considered affected by dysgraphia. The three scores are then normalized with normative mean and standard deviation. The global score is the median z-score: if this is less than the threshold -2, the pupil is considered affected by dysgraphia. The result of this test was therefore used to produce the dysgraphia risk labels used in this study, identifying 18% of the test population as “at-risk”. From now on, the five measurements taken at the five different time-points are indicated with the notation: y1, y2a, y2b, y3a, y3b as in Figure 2.

From the literature, it is assumed that the prevalence of dysgraphia is at least 5% [25], so in order to include at least 10 potentially pathological subjects in the sample, 200 children need to be enrolled. Taking into account the extended time frame that includes the transition from kindergarten to school, the sample size was increased to 250 kindergarten children to account for potential dropouts. Thus, the longitudinal study started with the inclusion of 247 children attending the last year of kindergarten. The acquisition was carried out directly in the selected schools, and in detail, in the locations in the province of Varese in Italy. Due to variable events beyond our control, as expected, some children were lost or added to the study at each measurement, resulting in a population of 210 children participating in the y3b acquisition and the BVSCO-2 Test. Both female (99) and male (111), right- (185) and left-handed (25) children were considered in the data collection. The sessions took place in quiet rooms, after ensuring that the participants were comfortably seated and in optimal lighting conditions. The invitation to participate in the study was extended to the entire class population, excluding only children who had previously been diagnosed with neurological or sensorimotor pathologies that could affect performance.

Overall, data were collected during the five time-points indicated by each participating child and for each of the proposed serious games. The experimental procedure was approved by the Ethics Committee of the Politecnico di Milano (no. 24/2019) and written consent was received from headmasters and parents.

### 3.2. Data Preprocessing

The app collected data from the interaction between the Apple Pencil and the tablet surface in terms of position (x and y coordinates), the pressure exerted, and the inclination of the pen. The data thus acquired were converted into multivariate time series composed of the positional features (x and y coordinates) and pressure. Features related to pencil inclination were discarded as they were on average constant and consequently uninformative in a classification problem. Since the sampling frequency of the Play-Draw-Write application in Unity was 50 Hz, the first step to obtain the time series was to sub-sample from the Apple Pencil, which instead samples at 240 Hz. At the same time, each time series was truncated by 5% of its length at the beginning and at the end to remove any insignificant sections and thus to avoid border effects. Children who did not participate in the final BVSCO-2 Test and whose labels were therefore not available were then discarded. The resulting data were normalized in x- and y-coordinates with respect to the initial position of the Apple Pencil, and the time vector was re-scaled in the range from zero to one.

Every serious game was played by the children in each of its modes, i.e., 3 executions for the two Copy Games and 15 executions for the two Tunnel Games, as summarized in Figure 3. To ensure an impartial assessment of the children’s manual dexterity, the order of execution of the Tunnel Games was always kept random from child to child. In order to facilitate a robust and stable analysis, the executions of the Tunnel Games are grouped according to the Index of Difficulty (ID), defined as the ratio between the length and width of the tunnel considered. Specifically, the 15 executions of Tunnel Games are such that they can be grouped into 5 IDs for the Tunnel Square and 8 IDs for the Tunnel Word. In order to group the randomly ordered Tunnel Games executions and to obtain a unique and more stable signal for each ID that is comparable between the children, the average signal was thus chosen as the representative execution for that ID, fixing measurement, game, feature, and child. From now on, for ease of notation, the term “ID” will be used instead of “copy mode” also for Copy Games.

Up to this point, the features available for each game consist of raw multivariate time series to which simple cleaning operations, i.e., cutting off heads and tails and normalization, have been applied. No explicit computation of high-level features has been added, precisely to allow subsequent steps to learn how to extract the most important information in a fully automatic manner. Indeed, the next step was to apply a nonlinear transformation to the previously processed data in order to reduce their dimensionality and extract a more meaningful representation. For this purpose, an embedding technique called Time2Vec was chosen and used exploiting deep learning algorithms. This approach is related to time decomposition techniques that encode a time signal into a set of frequencies but, instead of using a fixed set of frequencies as in Fourier transforms, frequencies can be learned [26]. Given a generic time instant τ and a number of components *s*, the Time2Vec function is a vector of size *s* + 1 defined as:(1)$$t2v\left(\tau \right)\left[i\right]=\left\{\begin{array}{cc} \omega_{i}\tau +\varphi_{i}, & \;i=0\\ F\left(\omega_{i}\tau +\varphi_{i}\right), & \;1\leq i\leq s \end{array}\right.$$
where *F* is a periodic activation function, in this case the sine one, and $\omega_{i}$ and $\varphi_{i}$ are learnable parameters.

An ad hoc neural network belonging to the autoencoder family was constructed for each feature of each collected sample. Such architectures, composed of two sequential sub-networks, namely the encoder and the decoder, are particularly convenient precisely for creating latent, low-dimensional representations of the input data [27]. The idea behind such a model lies in the quality of reconstruction of the input after passing through the bottleneck represented by the latent space: the higher the quality of the reconstructed data, the higher the quality of its compression. In this case, the architecture identified as optimal involves a fully connected layer of the same dimensionality as the input as encoder, a latent space consisting of a Time2Vec layer of tunable dimension, and a fully connected layer again of the same input dimensionality as decoder for reconstructing the data.

For each autoencoder, three vectors of learned parameters have been considered, as shown in Figure 4. In detail, the two vectors of the latent representation, i.e., pulsations $\omega_{i}$ and phases $\varphi_{i}$ with *i* = 1, …, *s*, and the vector of weights of the decoder’s layer, representing the importance of each of the sinusoids found by the network, i.e., $p_{i}$ with *i* = 1, …, *s*. Since the most discriminative features for the identification of handwriting difficulties are those related to frequency [11], the linear component of Time2Vec, i.e., the component corresponding to *i* = 0, was excluded from the extracted feature set and used only at the reconstruction stage.

Given the heterogeneous size of the individual runs, each sample was divided into ordered windows consisting of 50 timestamps each. Since these windows were highly overlapping, they were selected with hops of 5 timestamps. To select the optimal number of Time2Vec components, the first 75% of each dataset was used as the training set and the remaining 25% as the test set of a grid search. The best reconstruction result was obtained by having the number of components *s* = 9, achieving on average a mean squared error between input and output in the order of magnitude of $10^{-6}$. Figure 5 shows an example of a test set window for the pressure feature extracted from the execution of one of the children with respect to the y1 Copy Square game. Because of this excellent embedding, each signal of hundreds of time instants has been synthesized into a compressed representation consisting of 3 vectors of length equal to 9.

### 3.3. Procrustes Analysis and Ablation Study

With the aim of introducing a distance metric between the obtained periodic signals, and thus between children’s signals, the technique of Procrustes Analysis was relied upon. Procrustes Analysis is a rigid shape analysis that uses similarity transformations, i.e., isomorphic scaling, translation and rotation, to find the best fit between two reference shapes. In other words, the goal of this technique is to make two input figures appear as similar as possible in order to compute a distance between them [28,29].

At this point in the data processing, the most intuitive choice would be to apply Procrustes Analysis to the aggregation of the 9 sinusoids obtained from the Time2Vec embedding. Two possible configurations were selected to perform this aggregation, configurations that became part of the tuning hyperparameters of this ablation study phase. In detail, the choices evaluated consist of:Using the average of the sinusoids weighted with respect to the absolute value of the decoder weights. In this way, the resulting sinusoids combination is bounded in the range −1 and 1.Using the sum of the sinusoids weighted with respect to the real value of the decoder weights. In this way, the resulting sinusoids combination would not necessarily be contained between −1 and 1.

At this stage of data processing, the Procrustes Analysis was applied to periodic signals obtained as combinations of *s* sinusoids. Regardless of the configuration of the aggregation, the Procrustes Analysis needs reference landmarks in both sinusoids to be performed. In this context, the natural choice was to consider the extremes of the curves obtained. Since this technique requires the number of reference landmarks on the two shapes analyzed to be the same, the N extremes furthest from the x-axis were considered, with the hyperparameter N dynamically tuned. Once the landmarks on two different periodic signals had been identified, the two possible choices for pairing them were the following, as shown in Figure 6:To associate the landmarks according to their distance from the x-axis, and thus according to an order by value.To reorder the landmarks according to the time coordinate and thus to associate them in order of appearance.

Once the matching criterion has been chosen via tuning, it is possible to exploit it for each measurement and each game to construct very simple dysgraphia classifiers based solely on the threshold and the metric resulting from the Procrustes Analysis. In detail, the training technique involves for each ID and feature the calculation of the average signal among all children and the subsequent application of the algorithm from the Procrustes Analysis between the children’s signals and the average signal. The distances thus obtained are used for the calculation of the optimal threshold for each ID and feature, with respect to the BVSCO-2 Test score. Finally, a voting on IDs and features classifiers is performed, assigning to each child a risk probability. Again, it is possible to calculate the optimal threshold, but this time, a probability threshold. Algorithm 1 provides a concise representation of the described technique.
**Algorithm 1** Fit Procedure. 1: **for** eachID **do** 2:   **for** eachfeature **do** 3:    Computation of the average signal among all children 4:    Landmarks selection on the average signal 5:   Computation of children average signal distances via Procrustes Analysis 6:    Research of the best distance threshold to classify each child as “at-risk” or “not-at-risk” 7:   **end for**  8: **end for** 9: Voting on IDs and features to assign a dysgraphia-risk probability to each child10: Research of the best probability threshold to classify each child into one of the two classes

The output of the Fit Procedure is thus a classifier that assigns each child a probability corresponding to the risk of dysgraphia. The choice of how to optimize the threshold in turn includes the selection of a metric appropriate to the problem. Although Accuracy is often considered the best choice in classification problems, in this case, the strong class imbalance requires a metric that takes into account the distribution of labels. For this reason, the F1-score was chosen between Precision and Recall, i.e., their harmonic mean.

Once the implementation strategy of the classifiers, the hyperparameters to be tuned and the evaluation metrics have been defined, the last necessary step is the choice of the criterion to validate and compare the performance obtained. Although there are computationally much less onerous techniques, the technique of stratified nested k-fold cross-validation was preferred. This technique consists of a double loop, an external one for test sets and an internal one for validation sets, and thus allows an extremely robust and objective evaluation of the entire dataset at hand. In the conducted experiments, a k = 5 was chosen, so as to obtain 5 external folds and 5 internal folds, as illustrated in Figure 7.

The implemented version involves overlapping internal folds in order to keep an adequate size of each validation and test set, even given the limited cardinality of the dataset. The in-depth description of the entire pipeline is summarized in Algorithm 2. In detail, for each iteration of the algorithm, the possible configurations are evaluated, i.e., whether to aggregate the 9 sinusoids by averaging with absolute weights or with weighted sum, and whether to associate the landmarks according to the values of the obtained curves or following the temporal ordering. The number of landmarks varies dynamically from two, i.e., the minimum number of points to perform the Procrustes Analysis, up to the maximum value of comparable extremes between the curves, depending on both the measurement and the game. At the end of each internal cycle, the best configuration is selected on the basis of the average F1-score between the validation sets, and the performance on the test set is finally evaluated.

The final result is then a double validation performance average and a single test performance average for each of the four games and each of the five time-points considered. Since these are very simple classifiers, it was opted to construct a single model for each time-point that adaptively considers the contribution of each model associated with each game. A model ensembling technique called Blending and particularly in vogue in the world of machine learning competitions was therefore opted for. In particular, it is a meta-model that learns how to combine the predictions of simple models in an optimal manner, resulting in a robust and high-performance classification algorithm [30,31].
**Algorithm 2** Stratified Nested 5-fold Cross-Validation. 1: **for** eachouterfold **do** 2:   **for** eachinnerfold **do** 3:    **for** eachpossibleconfiguration **do** 4:     **for** eachID **do** 5:      **for** eachfeature **do** 6:        **for** eachchild **do** 7:         Computation of sinusoids combination to obtain a unique periodic representative signal 8:        **end for** 9:      **end for**10:     **end for**11:     **for** eachpossiblenumberoflandmarks **do**12:      Fit Procedure on training set13:     **end for**14:     Selection of the best number of landmarks on the current validation set15:    **end for**16:   **end for**17:  Selection of the best configuration based on average performance on the inner validation folds18:  Voting on current test set using the 5 classifiers probabilities from the best configuration to obtain each dysgraphia-risk probability19:   Final classification on current test set, using as threshold the average probability threshold from the 5 classifiers20: **end for**21: Concatenation of all the test predictions and performances evaluation on the whole dataset, in micro averaging fashion

The meta-model created in this work was then implemented for each of the five acquisition time-points, using in each of them the outputs produced by the four classifiers associated with the four games, as shown in Figure 8. It was thus possible to obtain two datasets, one extracted from the 25 validation folds and used to train and evaluate the meta-model (from now on named meta-training set), and one extracted from the 5 test folds and used in turn as the final test set (from now on named meta-test set), in both cases consisting of four features corresponding to the predicted probability.

A Quasi-SVM was chosen as the meta-model [32]. It is a small neural network composed of a single hidden layer called RandomFourierFeatures with a tuned unit number and a single output neuron with sigmoidal activation aimed at simulating the behavior of a Support Vector Machine [33]. Specifically, such a model was trained with the hold-out technique on the meta-training set divided in a stratified manner into 80% and 20% portions and finally tested on the meta-test set. The training procedure was performed for 1000 epochs minimizing Hinge loss with the Adam optimizer, learning rate equal to $10^{-4}$, early stopping with patience equal to 50, and the class balancing technique [34].

## 4. Results

In this section, results related to the classifiers, their configurations, and the realized meta-models are presented. Finally, a quantitative comparison in longitudinal perspective is made and the predictions of the best realized algorithm are inspected.

All choices regarding threshold-based classifier configurations were made with respect to validation performance within the stratified 5-fold cross-validation. The same configuration is then reapplied on the model to be evaluated on the test set. This means that for each time-point and for each game, there are 5 best configurations. Since there are 5 time-points and 4 games, the choice of the best configuration is repeated 100 times in total. The results in terms of best configurations are reported below:Weighted average of sinusoids and landmarks aggregated by value order: 26%.Weighted average of sinusoids and landmarks aggregated by time order: 27%.Weighted sum of sinusoids and landmarks aggregated by value order: 25%.Weighted sum of sinusoids and landmarks aggregated by time order: 22%.

It is noticeable how, despite the slight overall percentage variation, there is no particularly dominant configuration over the others, both from a general point of view and with regard to the aggregation of sinusoids and sorting for Procrustes Analysis. The search algorithm therefore managed to exploit each configuration almost equally, demonstrating their effectiveness depending on the context.

The optimal number of landmarks obtained as a result of the tuning step varied greatly from game to game, and, in general, from run to run, as is evidenced by the average number of points identified equal to 51.47 and its standard deviation of 42.80. In Figure 9, it can also be seen that the models based on the Tunnel Word game required fewer points than the other games, they also demonstrate the effectiveness of tuning the N parameter despite the increase in computation time.

Regardless of the configuration used, the threshold-based classifiers were used to make the prediction vectors, which were exploited to calculate the metrics and construct the meta-models. In addition to the models presented, a dummy-classifier called Baseline was added, which classifies each child as “not-at-risk”, i.e., consistently predicts the most likely class. The first set of results can be found in Table 1, which shows the main metrics used to assess the quality of classification models, regardless of cardinality and class balance. Table 2, on the other hand, shows the second set of results containing the most informative metrics in problems with unbalanced classes, and taking into account the proportions of classes within the various datasets.

For time-point y1, in Table 1, it is possible to notice that the Quasi-SVM classifier performs better in all metrics than the Baseline and, in general, can be considered the best model. Accuracy, Precision, and Specificity are much higher than the other threshold-based classifiers, while Recall and F1-score are slightly lower than the highest value, but the difference is very tiny. Precision and Specificity are both 100% with an overall Accuracy of 84.62%. The Copy Square-based classifier performs slightly better than the Quasi-SVM in terms of Recall and F1, but this result is counterbalanced by an increase in the false positive rate, which drops Precision to 18.75% and at the same time reduces Accuracy to 78.22%, which is clearly below the Baseline. The fact that the Quasi-SVM classifier is the best can be seen even better from Table 2: all weighted average metrics are much higher than both the Baseline and the single-game classifiers.

For time-point y2a, Table 2 shows that the Quasi-SVM classifier is the best among the other classifiers regardless of the considered metric. Accuracy is equal to 82.59%, Precision to 100% and, consequently, Specificity to 100% (Table 1). As far as Recall and F1-score are concerned, these are higher in the Tunnel Word-based classifier which, however, having a very low Precision and slightly lower Accuracy and Specificity, can be considered overall worse than the Quasi-SVM classifier.

For time-point y2b, the Copy Sequence-based model achieves the best weighted averages of Recall and F1-score (Table 2), but the slight improvement with respect to the Quasi-SVM model cannot compensate the loss in weighted Precision. From Table 1, the meta-model continues to have a perfect Precision, much higher than other models. Furthermore, despite the Copy Sequence Classifier achieves higher Accuracy, the difference is negligible, since it is in the first decimal digit. The same applies to the other competitors, e.g., the Tunnel Word-based classifier, which detects more children “at-risk” and therefore has higher F1-score and Recall than the Quasi-SVM model, but these values are not sufficient when considering the substantial loss in Accuracy and Precision. However, it is interesting to note that the acquisitions made during time-point y2b succeed in being exploited positively by most of the proposed models with respect to the Baseline, as in the case of the Copy Sequence-based classifier, which even on its own produced good results.

For time-point y3a, in Table 1, it is possible to see that the Recall of Quasi-SVM increases slightly compared to the other measurements at the expense of Precision, which for the first time reaches the nonetheless competitive value of 66.67%. Specificity also decreases albeit slightly, reaching 98.82%. Here, the Copy Square-based model gets the best Recall and F1-score while the best Accuracy is still obtained by the Quasi-SVM model. Looking also at the weighted averages (Table 2), the meta-model is still the best choice.

Finally, for time-point y3b, the Quasi-SVM classifier collapses and behaves in the same way as the Baseline by predicting all children in the Test set as “not-at-risk” (Table 1 and Table 2). It is therefore necessary to look for an alternative, but although the Copy Square-based classifier gets the best scores for Accuracy, Recall, and F1-score, the actual values are not comparable with those obtained from the meta-models implemented for the previous time-points. This is confirmed also by the lower weighted averages of Recall and F1-score compared to the Baseline model.

### Longitudinal Analysis

From now on, the analysis of the results proceeds by considering only the Quasi-SVM model for all the time-points, since it has been consistently better on average than Baseline and the other single-game-based classifiers. In general, the meta-model achieves similar performances in the five measurements except for the last one. It is possible to visualize the phenomenon described in Figure 10, where a picture of the evolution of the weighted averages of the metrics over time is shown, comparing the Baseline model and the Quasi-SVM.

The rational explanation behind why the meta-model obtains the best scores in the first measurement, i.e., the one most temporally distant from the collection of labels by BVSCO-2 Tests, is to be found in the temporal variation in the average level of children’s cognitive and motor skills. In our opinion, it is indeed very likely that the discrepancies between drawing skills and writing skills may have changed at different rates, and that this results in Play-Draw-Write game executions with patterns more similar to those captured by the BVSCO-2 Test precisely during the last year of kindergarten. Another reason identified lies in the fact that as children approach the second year of elementary school, the apparent differences between them in passing the proposed serious games are smaller, resulting in a more complex classification problem. In addition to its performance, it is for these reasons that the Quasi-SVM was chosen as the final dysgraphia classifier to be proposed from February of the last year of nursery school, i.e., starting well over 2 years earlier than current diagnosis protocols.

Figure 11 shows the BVSCO-2 z-score obtained from the children participating in the study. The dotted line indicates the classification threshold of the test, while the colored dots are the children classified as “at-risk” by the classifier. It can be seen that the children identified over time by the Quasi-SVM model are different at each time point, except for a single child who is identified at two different time-points, namely y1 and y3a. With the exception of precisely time-point y3a, for which the model has found an optimal class separation hyperplane that however predicts the presence of false positives, all other models manage to identify at least one child as correctly “at-risk” with the highest possible Accuracy. This turns out to be an advantage for two reasons: on the one hand, the moment the algorithm presents a report of dysgraphia during the last year of kindergarten and the first year of primary school, it can be given full confidence. On the other hand, the children identified as dysgraphic are not those with the worst z-scores, which would be easily detected by a teacher or clinician, but those closer to the threshold and probably with characteristics less distinguishable from the “non-at-risk” population. Therefore, thanks to the developed model, Play-Draw-Write detects children whose difficulty is not so evident yet, but that will emerge over time.

## 5. Discussion

The awareness that Dysgraphia is a very complex phenomenon is increasing [4], as well as the awareness that its classical diagnosis has several weaknesses. The latter, in fact, is delayed since it is based on the assessment of the speed and quality of the final handwritten text, when handwriting is mastered: it thus relies on a subjective evaluation of the final product. Moreover, it can also be further delayed by an often expensive and difficult to be reached professional consultation [10].

With this in mind, this work focused on the possibility of digitizing the early screening of dysgraphia by exploiting the iPad application Play-Draw-Write. Thanks to the acquisitions carried out, the reduction of their dimensionality by means of deep learning techniques, the realization of classification models based on Procrustes Analysis and the ensemble technique used to realize the final meta-models, it was possible to confirm that symbol drawing allows for the objective assessment of handwriting development. Symbol drawing has the advantage of being able to be carried out by people from any country in the world, thus enabling the identification of potential weaknesses in handwriting through an accessible and inexpensive technological support.

The one presented turns out to be an innovative approach in the literature and is evaluated over a wide time range precisely to identify the best moment to detect any delays in handwriting learning and intervene even before the consequences of such difficulty become apparent. From the encouraging results, there is evidence that such a preventive analysis is not only valid, but also finds its maximum effectiveness precisely in the instant of time most distant from that commonly chosen to assess children’s handwriting skills. It is also interesting to underline that the task, which is highly nonlinear, manages to be accomplished successfully and with performances that are far from obvious despite the scarcity of available data. Indeed, we are convinced that each model associated with the individual serious game as well as each meta-model can benefit in terms of performance and generalization from a larger number of children involved in data acquisition.

Regarding the performance obtained by the Quasi-SVM classifiers, it was deliberately preferred to opt for models that would penalize the Recall metric over Precision. Indeed, the goal was not to make a tool that would replace the role of a clinician rather than a teacher, but instead to support their work with a tool that would identify the “at-risk” class as accurately as possible. In this way, in addition to having a tool for confirming diagnoses with a zero false-positive rate, it is possible to identify precisely those individuals whose delays in learning to write turn out to be less obvious, ideally prioritizing them for a more extensive clinical analysis.

Thanks to the repeated use of the Play-Draw-Write application over time, it was possible to observe how the features extracted from the proposed exercises became less informative as the months passed and the children’s writing skills improved. This suggests that a possible future development could concern the acquisition of data at a time even before the last year of kindergarten, when the children’s ability to solve the drawing exercises also shows substantial differences. We argue that the use of the Play-Draw-Write application coupled with the proposed classification protocol may therefore be a successful tool in a pre-clinical setting, such as school or home, increasing the accessibility of early detection of handwriting difficulties. It is also important to emphasize the accessibility of such a tool as well as its acceptability by the children, who, being comfortable in a familiar environment and using a tool as familiar to them as the tablet, expressed a perceptible enjoyment.

Future developments of this work could focus on adding variables calculated from the raw signals, such as distance to a fixed point or instantaneous speed. Another option is to realize the autoencoder based on Time2Vec in a multivariate manner, thus considering information from the entire set of variables, and perhaps also to consider the linear (non-periodic) term of Time2Vec. Finally, other strategies can be developed to select landmarks on periodic signals, e.g., considering other references instead of the distance to the x-axis. With this work, we hope to have laid the roots for the emergence of a useful dysgraphia support for children, to help them build a better foundation for their learning and, consequently, for their entire lives.

## Figures and Tables

**Figure 1 life-13-00598-f001:**
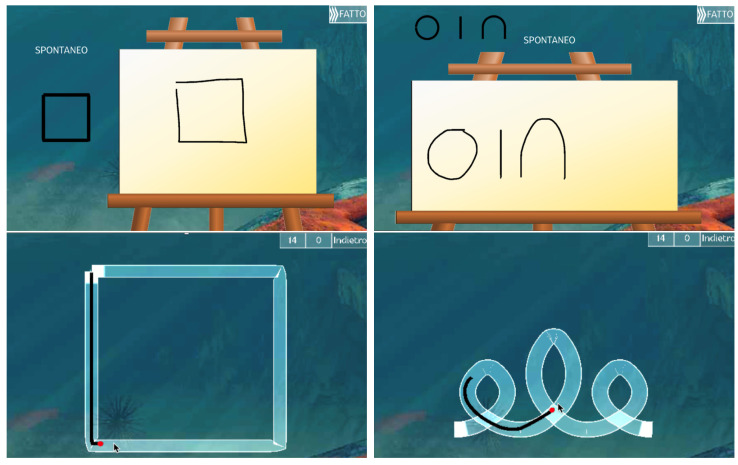
Screenshots of the games used for data collection in this paper. In the top row, from left to right, an example of Copy Square and Copy Sequence. In the bottom row, from left to right, an example of Tunnel Square and Tunnel Word.

**Figure 2 life-13-00598-f002:**
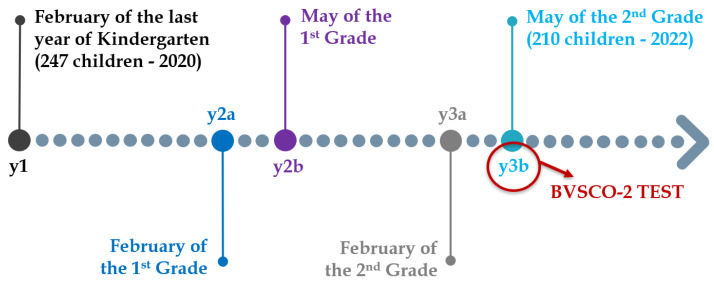
The timeline of the longitudinal study on dysgraphia carried out for this paper. The five data collections are spread over a three-year time window, at the end of which the BVSCO-2 Test has been carried out.

**Figure 3 life-13-00598-f003:**
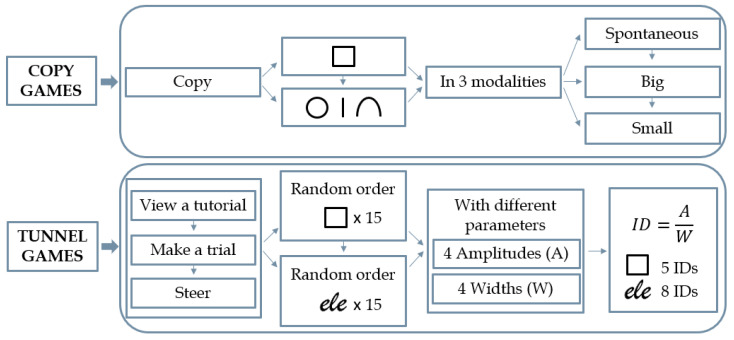
The study protocol for dysgraphia screening adopted at each measurement. Notice that the three executions of the Copy Games coincide with the three required copy modalities, while the fifteen executions of the Tunnel Games can be grouped according to the Index of Difficulty (ID).

**Figure 4 life-13-00598-f004:**
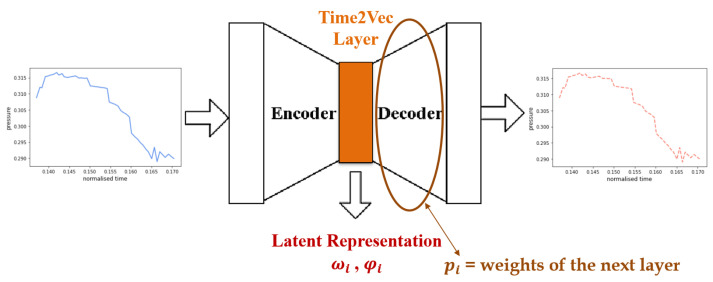
The diagram of the autoencoder architecture used in this work. The data are compressed by the Encoder within the latent space formed by a Time2Vec layer, and then reconstructed by the Decoder as accurately as possible.

**Figure 5 life-13-00598-f005:**
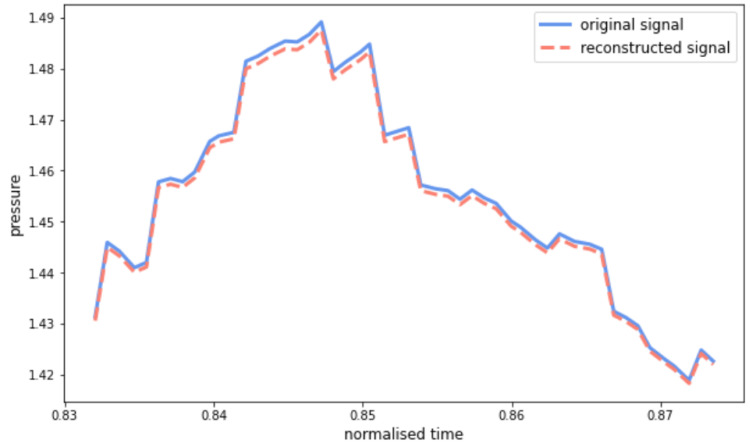
An example of pressure signal reconstruction of a sample belonging to the test set. It is possible to notice how the excellent analytical performance is reflected in a high-quality reconstruction.

**Figure 6 life-13-00598-f006:**
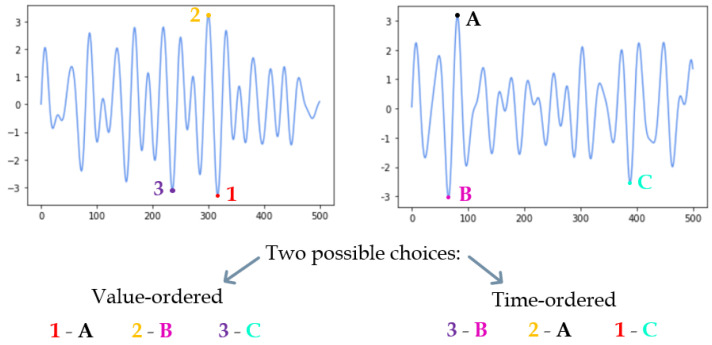
A toy example of landmark-matching for Procrustes Analysis in the time series domain with N = 3. The choices considered involve sorting by curve values, or sorting by time order.

**Figure 7 life-13-00598-f007:**
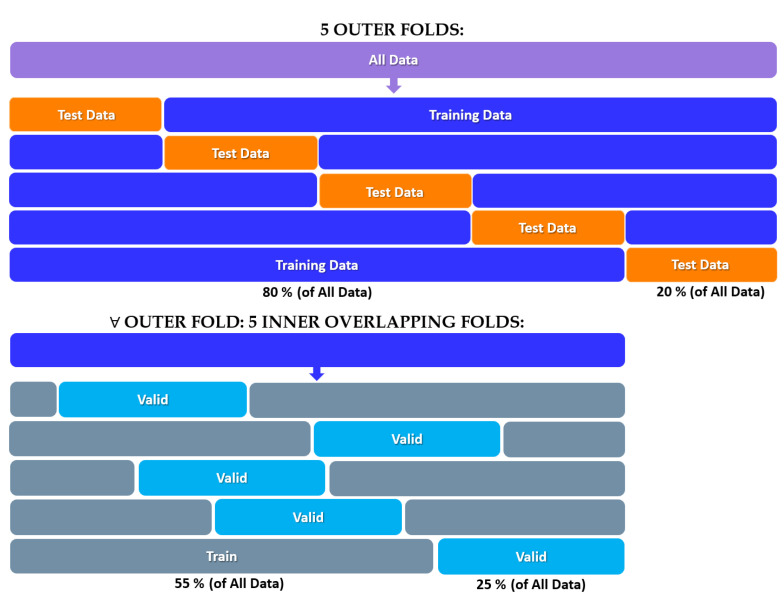
The diagram of the Stratified Nested 5-fold Cross-Validation strategy used to evaluate the performance of the proposed models.

**Figure 8 life-13-00598-f008:**
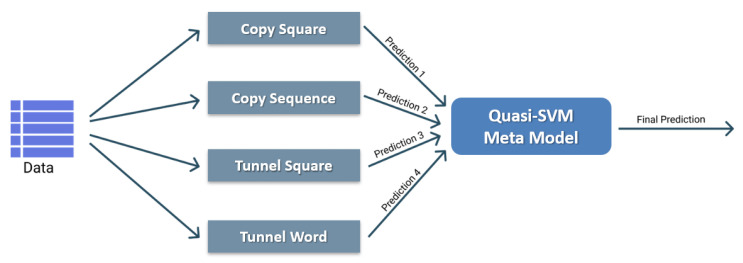
Schematic representation of the meta-model built from the predictions of the four threshold-based classifiers.

**Figure 9 life-13-00598-f009:**
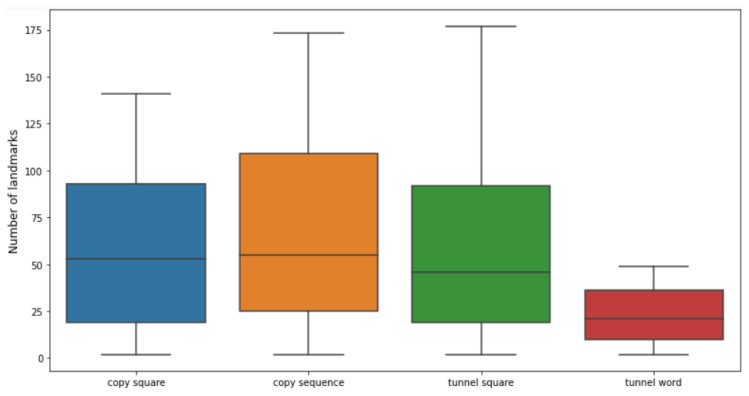
The optimal number of landmarks identified by the tuning performed via stratified nested 5-fold cross-validation, divided by each game and represented as a boxplot.

**Figure 10 life-13-00598-f010:**
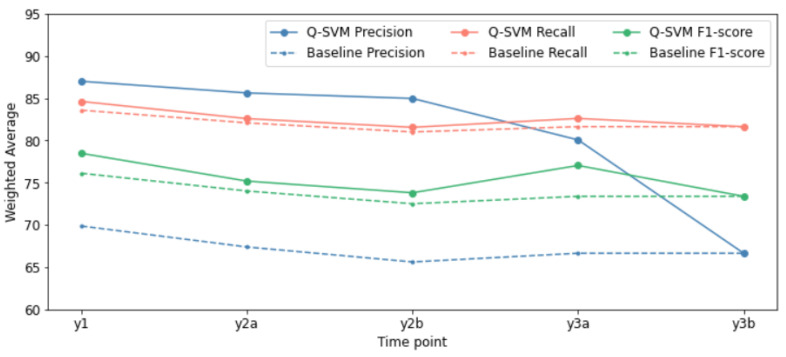
A summary of the weighted averages of the main metrics considered to compare the models presented. For each time-point, the Quasi-SVM meta-model is compared with the Baseline.

**Figure 11 life-13-00598-f011:**
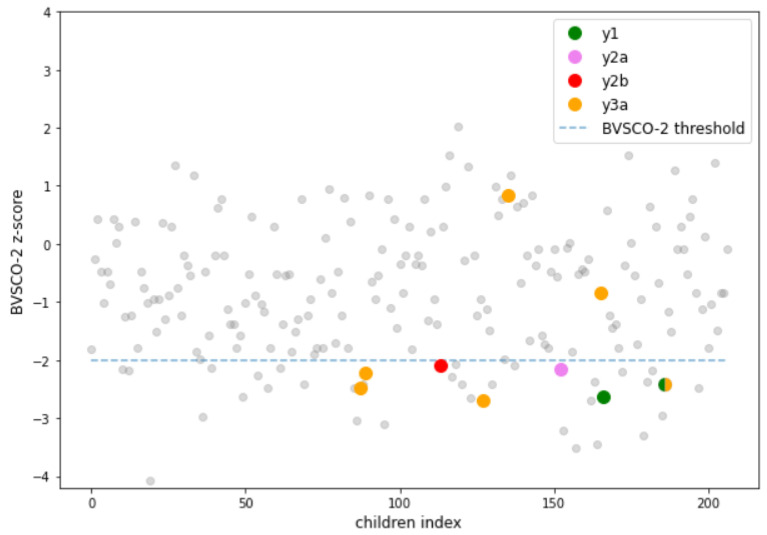
Scatterplot of the z-score of the BVSCO-2 test obtained by the children participating in the study. The colored dots, below the threshold, represent the subjects predicted as “at-risk” by the Quasi-SVM classifier.

**Table 1 life-13-00598-t001:** The metrics calculated on the respective test sets used to compare the classifiers realized for each time-point. Values in bold represent the best results, underlined values the second best.

Data	Model	Accuracy	Precision	Recall	Specificity	F1-Score
**y1**	Baseline	83.59	0.00	0.00	**100.00**	0.00
	Copy Square	78.22	18.75	**8.82**	92.26	**12.00**
	Copy Sequence	78.39	9.09	2.94	93.94	4.44
	Tunnel Square	79.31	0.00	0.00	95.83	0.00
	Tunnel Word	80.40	14.29	2.94	96.36	4.88
	Quasi-SVM	**84.62**	**100.00**	6.25	**100.00**	11.76
**y2a**	Baseline	82.09	0.00	0.00	**100.00**	0.00
	Copy Square	81.64	25.00	2.78	98.25	5.00
	Copy Sequence	77.78	15.38	5.41	93.53	8.00
	Tunnel Square	79.51	14.29	2.70	96.43	4.55
	Tunnel Word	72.33	11.54	**8.11**	86.39	**9.52**
	Quasi-SVM	**82.59**	**100.00**	2.78	**100.00**	5.41
**y2b**	Baseline	81.01	0.00	0.00	**100.00**	0.00
	Copy Square	78.95	11.11	2.70	95.35	4.35
	Copy Sequence	**81.82**	42.86	8.11	97.67	13.64
	Tunnel Square	81.63	33.33	2.86	98.76	5.26
	Tunnel Word	73.63	18.18	**11.76**	87.84	**14.29**
	Quasi-SVM	81.56	**100.00**	2.94	**100.00**	5.71
**y3a**	Baseline	81.64	0.00	0.00	**100.00**	0.00
	Copy Square	78.10	33.33	**21.05**	90.70	**25.81**
	Copy Sequence	76.19	25.00	15.79	89.53	19.35
	Tunnel Square	76.81	25.00	13.16	91.12	17.24
	Tunnel Word	80.48	20.00	2.63	97.67	4.65
	Quasi-SVM	**82.61**	**66.67**	10.53	98.82	18.18
**y3b**	Baseline	**81.64**	0.00	0.00	**100.00**	0.00
	Copy Square	76.19	**16.67**	**7.90**	91.28	**10.71**
	Copy Sequence	78.10	10.00	2.63	94.77	4.17
	Tunnel Square	75.85	7.14	2.63	92.31	3.85
	Tunnel Word	74.29	13.64	**7.90**	88.95	10.00
	Quasi-SVM	**81.64**	0.00	0.00	**100.00**	0.00

**Table 2 life-13-00598-t002:** The weighted average metrics calculated on the respective test sets used to compare the classifiers realized for each time-point. Values in bold represent the best results, underlined values the second best.

Data	Model	Precision (WAvg)	Recall (WAvg)	F1-Score (WAvg)
**y1**	Baseline	69.87	83.59	76.12
	Copy Square	72.46	78.22	74.85
	Copy Sequence	69.91	78.39	73.57
	Tunnel Square	67.98	79.31	73.21
	Tunnel Word	71.10	80.40	74.69
	Quasi-SVM	**87.01**	**84.62**	**78.48**
**y2a**	Baseline	67.39	82.09	74.02
	Copy Square	72.71	81.64	75.08
	Copy Sequence	70.06	77.78	73.18
	Tunnel Square	69.63	79.51	73.37
	Tunnel Word	68.62	72.33	70.35
	Quasi-SVM	**85.63**	**82.59**	**75.19**
**y2b**	Baseline	65.62	81.01	72.51
	Copy Square	69.45	78.95	73.33
	Copy Sequence	76.03	**81.82**	**76.35**
	Tunnel Square	73.62	81.63	74.73
	Tunnel Word	69.47	73.63	71.31
	Quasi-SVM	**84.98**	81.56	73.81
**y3a**	Baseline	66.65	81.64	73.39
	Copy Square	74.73	78.10	76.05
	Copy Sequence	72.34	76.19	73.97
	Tunnel Square	71.82	76.81	73.80
	Tunnel Word	70.74	80.48	73.84
	Quasi-SVM	**80.07**	**82.61**	**77.04**
**y3b**	Baseline	66.65	**81.64**	**73.39**
	Copy Square	**69.99**	76.19	72.59
	Copy Sequence	68.56	78.10	72.53
	Tunnel Square	67.30	75.85	71.07
	Tunnel Word	69.12	74.29	71.43
	Quasi-SVM	66.65	**81.64**	**73.39**

## Data Availability

The data that support the findings in this study are available from the corresponding author upon reasonable request.

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
