# Peer review of "Deep Learning and Procrustes Analysis for Early Dysgraphia Risk Detection with a Tablet Application"

_life, 2023, doi:10.3390/life13030598_

Round 1
Reviewer 1 Report
The paper is about interestinf and important topic of early diagnosis of dysgraphia using machine learning methods.
The literature review is very superficial. The topic has been actively developed in recent years, it is completely incomprehensible why the author presented 2.1 only 4 studies, not all of which are directly related to the author's research. A simple ScholarGoogle search turns up hundreds of studies over the past two years.
The study itself looks quite reasonable and scientifically presented. However, in connection with the above problem with the analysis of the available literature, it is necessary not only to present the results, but to emphasize what is new in this paper in comparison with others.
Data on the children participating in the experiment should be presented more widely, not only their number.
For readers, it would be more clear if you presented screenshots of the game used.
Author Response
Dear Reviewer,
First of all, I would like to thank you on behalf of all the authors for your valuable comments, which have certainly helped to make this work more complete and coherent.
I will now turn to the timely responses.
We have expanded the section on related works, doubling the number of works cited and adding a final reference to the 2022 reference survey.
We are sure that readers will see the added value of our work thanks to these additions, just as you suggested.
We have expanded the section on information about the children recruited for the data collection, explaining the rationale for the number of subjects and specifying the inclusion and exclusion criteria.
Finally, as requested, we have added pictures of the games used for data collection, with appropriate citations in the text.
I would like to thank you again on behalf of all the authors for taking the time to review our paper and considering it fit for publication.
Reviewer 2 Report
Overall, this is a well-written manuscript. The authors showed a very detailed background and introduction. The listed algorithms are easy to follow and understand. The results session is well presented and depicted. I don't have any concerns to publish this article.
Author Response
Dear Reviewer,
I would like to thank you on behalf of all the authors for taking the time to review our paper and finding it suitable for publication.
Reviewer 3 Report
In this manuscript, the authors present the results of a longitudinal study based on the monitoring and detection of dysgraphia in children with machine learning models. The work appears scientifically sound, and the results obtained encourage the development of a system for the early screening of handwriting difficulties. While the paper is mainly well-structured and written, I have a few suggestions. First, it should be reread, as I encountered some typos, such as "realize realize", "label_s distribution", "k-fold_s", "e.g., considering for example". Second, the resolution of some figures could be better. Also, there is a problem with an arrow in Fig. 2. From a methodological perspective, more details regarding the handwriting tasks performed by the participants and the features sampled by the acquisition device should be added. Finally, other relevant recent literature should be discussed (see, for example, the recent survey [10.1007/s00521-022-07185-6]).
Author Response
Dear Reviewer,
First of all, I would like to thank you on behalf of all the authors for your valuable comments, which have certainly helped to make this work more complete and coherent.
I will now turn to the timely responses.
We have made the grammatical changes you kindly identified, and subsequently reviewed the entire work using language correction tools.
We then improved the quality of the uploaded images, as requested, and adjusted the arrow in Figure 2, as you pointed out.
We also expanded the section on the data collected and the method by which the features were first computed and then automatically revised using a deep learning algorithm.
Finally, as you requested, we have greatly expanded the literature described in the Related Works section, doubling the number of works cited and adding a reference to the survey you kindly suggested.
I would like to thank you again on behalf of all the authors for taking the time to review our paper and considering it fit for publication.